# Identity-Projection as a way of analyzing Attention Heads in Transformers

## Abstract

Transformer-based large language models (LLMs) exhibit complex emergent behaviors, yet their internal mechanisms remain poorly understood. Existing interpretability methods often rely on supervised probes or structural interventions such as pruning. We propose the notion of *identity-projection*, a property in tokens and prompts whereby the features they embed—directly or indirectly—reflect the same features they carry independently, even in different contexts. Leveraging the local linear separability of latent representations within LLM components, we introduce a method to identify influential attention heads by measuring the alignment and classification accuracy of hidden states relative to class prompts in each head's latent space. We find that these alignments directly affect model outputs, steering them towards distinct semantic directions based on the attention heads' activation patterns. In addition, we propose a novel unsupervised method, Head2Feat, which exploits this linear property to identify and align groups of datapoints with target classes, without relying on labeled data. Head2Feat is, to our knowledge, the first unsupervised approach to extract high-level semantic structures directly from LLM latent spaces. Our approach enables the identification of global geometric structures and emergent semantic directions, offering insights into the model's behavior while maintaining flexibility in the absence of task-specific fine-tuning.

## 1 Introduction

Transformer-based auto-regressive large language models (LLMs) (Vaswani et al., 2017), such as GPT (Brown et al., 2020) and LLaMA 3 (Grattafiori et al., 2024), have become the dominant architecture for natural language processing (NLP) tasks. Despite their success, the internal mechanisms that drive their behavior remain only partially understood. Their depth and complexity give rise to emergent abilities (Wei et al., 2022) that are difficult to isolate and analyze.

The Transformer architecture is composed of two primary components: a self-attention mechanism, which enables the model to read from previous tokens, and a multi-layer perceptron (MLP) block, which updates the current token representation (Elhage et al., 2021). Prior work has shown that attention heads (AH) can act in sequence to guide the MLP toward task-relevant features (Lv et al., 2024) , (Chughtai et al., 2024). This guidance emerges through the coordinated activity of multiple attention heads, which collectively shape the information passed to the MLP. Together, these components allow the model to integrate contextual information and generate coherent next-token predictions.

Attention heads have been shown to encode interpretable features such as truthfulness (Li et al., 2024), temporal structure, and geographical information (Gurnee & Tegmark, 2023). Remarkably, many of these properties can be recovered using simple linear probes (Alain & Bengio, 2016) applied directly to the output of individual attention heads. This suggests that LLMs often represent semantic features in a robust, locally linear way, similar to static word embeddings such as Word2Vec (Mikolov et al., 2013), but with the added flexibility of contextual adaptation.

Unlike the residual stream, which aggregates information into a shared latent space—facilitating the disentanglement of features—individual attention heads tend to operate in distinct subspaces and exhibit specialization in specific linguistic or semantic functions. Moreover, many attributes are distributed across multiple heads, rather than being localized to a single one. This dispersion complicates interpretability and undermines the effectiveness of simple linear probes for isolating and

localizing specific features. Yet, if these distributed representations align along consistent geometric directions, it may still be possible to uncover stable, interpretable features.

We introduce *identity-projection*, a property in tokens and prompts whereby the features they embed in any prompt—directly or indirectly—reflect the same features they carry independently. This property enables prompt classification without training, reveals attention heads with high attribution, and provides a simple mechanism to trace information flow within the model.

In this paper we make three contributions: (1) We demonstrate that tokens and prompts project their features into the subspaces of their parent promtps which we call *identity-projection*. (2) We introduce *IPA* (Identity Projection Analysis), a zero-shot method for classifying prompts and identifying the most important attention heads. (3) We propose *Head2Feat*, an unsupervised mechanism that clusters vectors according to their relevance to a given semantic feature.

## 2 RELATED WORKS

**The Geometry of Latent Space and Linear Representation** Works like Linear Representation Hypothesis (Park et al., 2023) showcase that hgih-level, abstract concepts reside in the latent space as linear directions within a model, which was extended via Frame Representation Hypothesis (Valois et al., 2024) which was generalized to various concepts. Other papers, such as Language Models Represent Space and Time (Gurnee & Tegmark, 2023) have discovered that models create a world model of concepts like space and time.

**Mechanistic Interpretability and Feature Discovery.** Understanding the internal workings of large language models (LLMs) is a significant challenge. One approach is using probes (Alain & Bengio, 2016) to assess if specific information is encoded in a model's hidden states. More advanced methods, such as autoencoders (standard (Hinton & Salakhutdinov, 2006), variational (Kingma et al., 2013), and sparse), aim to extract interpretable features from the latent space.

For transformers, techniques like Activation Patching (Meng et al., 2022) and Path Patching (Goldowsky-Dill et al., 2023) provide causal interpretability by identifying how specific behaviors or factual information are localized within the model. Tools like LogitLens (nostalgebraist, 2020) and TuneLens (Belrose et al., 2023) visualize and predict token probabilities. Recent work has also explored the roles of attention heads in managing knowledge conflicts (Jin et al., 2024) and automatic discovery of computational pathways (Kramár et al., 2024; Ferrando & Voita, 2024).

**Model Steering and Activation Engineering** A growing body of research focuses on manipulating LLMs by intervening in their internal activations. (Meng et al., 2022) introduced the Rank-One Model Editing (ROME) method, which allows for causal tracing and editing of factual associations in a model's feed-forward layers, establishing the localization of knowledge within model parameters.

Recent advances in activation engineering have enabled real-time manipulation during inference. Techniques like Inference-Time Intervention (ITI) (Li et al., 2024), Context-Aware Activation Addition (CAA) (Panickssery et al., 2023), In-Context Vectors Liu et al. (2023), and Style Vectors (Konen et al., 2024) can guide model outputs toward desired behaviors without retraining. Our proposed method of using self-representation for analysis and steering aligns with these approaches, offering a way to uncover and leverage influential semantic directions in an unsupervised manner.

## 3 IDENTITY-PROJECTION IN TOKENS

A central question in understanding LLMs is how semantic information is encoded and flows through the model. We argue that semantic attributes are encoded as consistent, high-dimensional directions within the model's representation space, which remain stable across different contexts and can be activated even when the associated token is not explicitly present.

Past research (Li et al., 2024; Gurnee & Tegmark, 2023; Konen et al., 2024) has shown that there exist directions in attention heads that are invariant to context. We define prototypes as these invariant semantic directions $\mathbf{p} \in \mathbb{R}^d$, which encode specific meaning (e.g., "France-location," "joy-sentiment," "past-tense"). The attribute subspace $S_a \in \mathbb{R}^d$ is the space where semantically similar prototypes reside.

We argue that this prototype invariance extends to tokens and prompts, where it is equivalent to their semantic identity. That is, part of the token's identity is encoded in the attention heads, which we refer to as *identity-projection*: when semantically related information is present in a context, the model activates the same prototype directions that would be activated if the token itself were explicitly mentioned. For example, processing "Emmanuel Macron" activates the same "France-location" prototype that would be activated by explicitly mentioning "France," enabling the model to maintain consistent geographical representations even when France is only implicitly referenced.

The following proposition formalizes this intuition:

**Proposition 1** (Shared Feature Directions via Identity-Projection). *Let $h_n : T \times C \to \mathbb{R}^d$ denote the representation function for attention head $n$ mapping token $t$ in context $C$ to a $d$-dimensional vector. Assume a conceptual semantic distance $d$ where $d_{sem}(t, s)$ is the shortest path length between tokens $t$ and $s$ in a semantic association graph. Given that active features are invariant within tokens, for token $t$ and any context $C'$, define:*

*1. Prototype vector: $p_t := h_n(t, t)$*

*2. Self-consistency: $\langle h(t, C'), p_t \rangle > 0$*

*3. Distance-decay: For token $s$ with $d_{sem}(t, s) = k$ and a decay function $f(k)$:*

$$\langle h_n(s, C'), p_t \rangle \geq f(k)\langle h_n(t, C'), p_t \rangle$$

*4. Orthogonality: For tokens $u$ with $d_{sem}(t, u) \to \infty$: $\langle h_n(u, C'), p_t \rangle \approx 0$*

Then, to quantify the influence of a prototype on a given prompt, we project the input vector $v$ onto the prototype vector $p$. The influence score is given by:

$$I = \frac{\langle v, \mathbf{p}_t \rangle}{\|\mathbf{p}_t\|}. \tag{1}$$

This score indicates how strongly the prototype is activated in a specific context, allowing us to assess the role of each prototype in the model's processing of a given prompt.

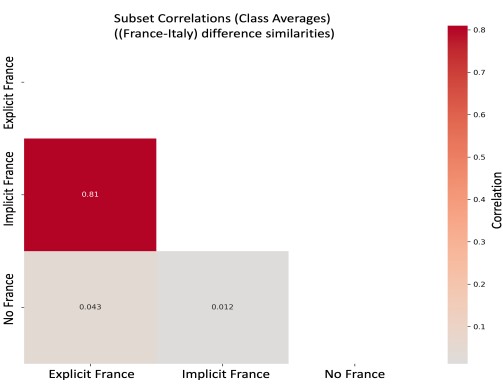

(a) Correlation between the subsets of prompts

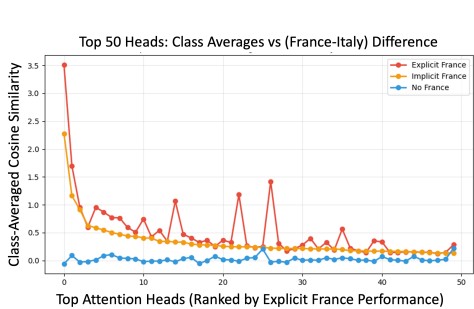

(b) Alignment of attention heads with respect to base tokens

Figure 1: Correlation between the difference of different sets including prompts with the word France/Italy, prompts related to France/Italy and unrelated prompts.

To empirically validate this property, we compare three levels of similarity with respect to a target token: explicit mentions, implicit references, and unrelated prompts. We expect a monotonic decrease in alignment across these three levels. To this end, we constructed six datasets—three for France and three for Italy—partitioned into explicit, implicit, and unrelated sets. For each group, we averaged results and computed differences along the France–Italy direction

$$\mathbf{p}_c = \frac{1}{P} \sum_{i \in P} v_i - \frac{1}{N} \sum_{i \in N} v_i \tag{2}$$

where $P$ and $N$ are sets of positive and negative examples, respectively. And then, quantify their magnitude using Eq. 1.

Figure 1b shows that the top-50 heads from the implicit set strongly align with those from the explicit set, while correlations with unrelated prompts remain near zero. The implicit–explicit correlation reaches 0.85 (Figure 1a), consistent with the Proposition 1 prediction that related tokens share feature directions and unrelated ones will have orthogonal views in it.

These properties enable us to analyze and interpret model behavior systematically. In particular, we can align token features with respect to prompts using two complementary approaches: aligning tokens between each other and aligning contrastive prompts with the respective contrastive token. This alignment allows us to quantify which components of the model carry specific semantic information and how different prompts activate these components.

### 3.1 ATTRIBUTION SCORING THROUGH SELF-REPRESENTATION

We leverage our *identity-projection* framework to identify which attention heads encode specific semantic prototypes. We present two complementary approaches which we both call IPA: multi-class classification using multiple token prototypes, and contrastive analysis using paired token differences.

**Multi-Class Token Classification** For semantic domains with multiple classes (e.g., languages, emotions, character styles), we extract a prototype for each class using its name token embedding as a reference direction. Given a prompt, we classify it by computing the influence score (Eq. 1) for each class prototype and selecting the class with the highest score.

To identify the most important attention heads for each semantic domain, we evaluate classification performance using F1-score across all heads. Heads that achieve high F1-scores are considered to strongly encode the corresponding information from that class.

Interestingly, all languages share important attention heads in layers 3, 16, 17, and 24, as seen in Figure 2a. Countries also share attention heads in layers 16 and 24 with languages, while emotion-related features appear to be concentrated in layer 3. Most character styles are embedded in the second half of the model, with the notable exception of Yoda, as shown in Figure 2b. All the heatmap graphs can be found in the Appendix

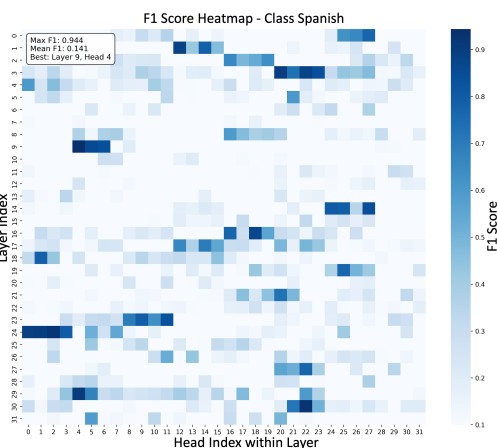
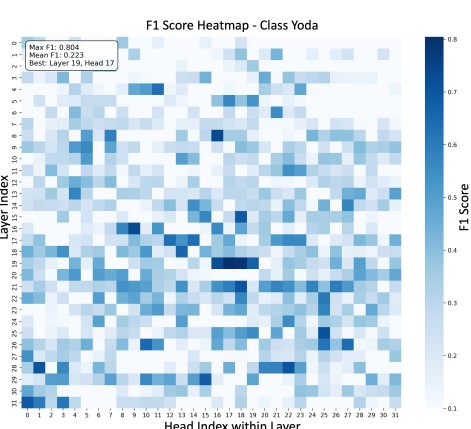

(a) Attention Head F1-Score based on Spanish sentences

(b) Attention Head F1-Score based on Yoda speaking style

Figure 2: Influence scores obtained from the dataset TruthfulQA with respect to the "Truthful" - "Untruthful" vector

**Contrastive token analysis** For binary semantic distinctions, we extract prototypes using contrastive token pairs. We compute the prototype direction using Eq. 2

We then rank attention heads by their influence scores when aligned with this contrastive prototype. Figure 3 compares two approaches:

- **Single pair**: Using one contrastive prompt pair to extract the prototype
- **Multiple pairs**: Averaging across 1,500 contrastive pairs from TruthfulQA (Lin et al., 2021)

Both approaches identify similar high-influence attention heads, with the top-5 heads showing significantly stronger prototype activation. The single-pair approach seems to deviate from the multi-pair one, but, for the truthful case it requires only a few critical heads while the rest are just needed as support. Interestingly, preventing output reversion to baseline requires patching several lower-influence heads as well.

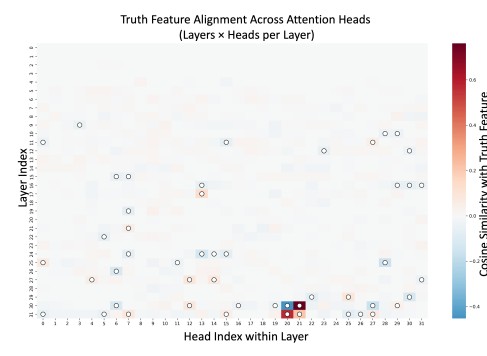

(a) A pair of prompts                    (b) 1500 pairs of prompts

Figure 3: Influence scores obtained from the dataset TruthfulQA with respect to the "Truthful" - "Untruthful" vector

### 3.2 ACTIVATION PATCHING

To validate that our identified attention heads actually control the semantic features, we employed activation patching (Zhang & Nanda, 2023) experiments. We used continuous activation patching where we modify specific attention head outputs by adding activations from a reference ("corrupted") prompt that exhibits the desired semantic property, then evaluate whether the model's behavior shifts accordingly.

We tested our method on multiple semantic domains identified in Section 3.9. For each target style, we:

- Selected the top-ranked attention heads based on F1-scores or influence scores from our attribution analysis
- Patched clean activations with corrupted prompts using the format "Answer as/in style" prepended to the initial prompts
- Generated 70 tokens for each patched prompt and evaluated the output

We primarily relied on attention heads with the highest attribution scores from our analysis. In some cases, we found that including several of the main heads associated with "English" improved performance, suggesting that effective semantic steering requires both activating the target style and suppressing the default (English) style.

The prompts used followed the format "Answer as/in style", which was prepended to the question prompt, and we generated 70 tokens for each. For selecting the attention heads, we primarily relied on those with the highest F1-score or Influence Score from Section 3.1. In some cases, adding attention heads from the "English" results helped improve the accuracy of the generated answers. This suggests that steering the output depends not only on attributing the desired style, but also on mitigating the influence of the style currently being used.

From Table 1, we observe that changing languages was highly effective, the number of attention heads changed was around 20 which is equivalent to 2% of the AH (for the rest of the elements the amount of heads had to be of 40 which is 4%) and the metrics showed good results. In contrast,

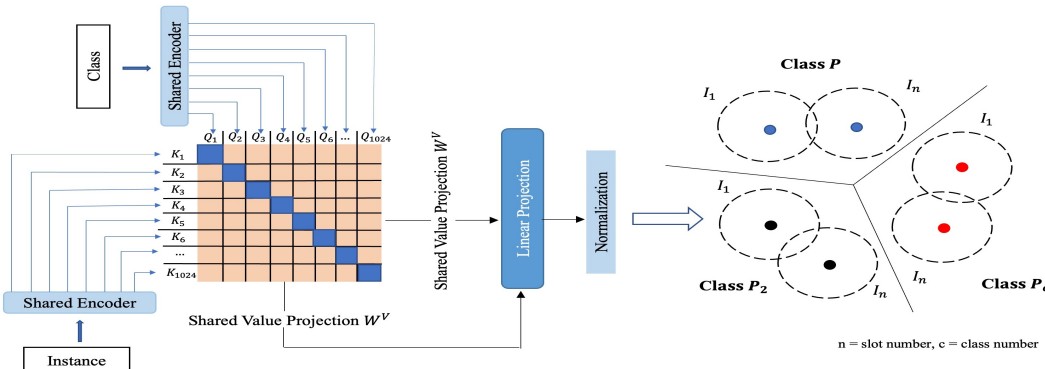

Figure 4: Overview of *Head2Feat* architecture. The model uses an attention mechanism to pair class and instance vectors projected into a shared attribute space, where prototypes are learned to identify attention heads encoding semantically relevant features related to the target attribute.

Table 1: Results from the activation patching using different styles

| Style | BERT-Spanish Score | BERT-German Score | Accuracy (%) | Emotion Classification |
|---|---|---|---|---|
| Spanish | 0.72 | 0.57 | 100 | Neutral |
| German | 0.60 | 0.66 | 99.0 | Neutral |
| Sad | 0.62 | 0.55 | 68.3 | Sadness |
| Angry | 0.59 | 0.57 | 59.4 | Anger |
| Truth | 0.61 | 0.57 | 100 | Neutral |
| Lie | 0.61 | 0.57 | 9.9 | Neutral |
| Random | 0.62 | 0.58 | 100 | Neutral |
| Default | 0.62 | 0.58 | 100 | Neutral |

altering the emotional style improved emotion classification but reduced factual accuracy, as the model prioritized emotional expression over precise question-answering.

These results confirm that our attribution method successfully identifies attention heads that have high attribution values towards specific semantic properties, towards the point of only needing 4 prompts to obtain the scores

### 3.3 UNSUPERVISED CLASSIFICATION THROUGH PROTOTYPE ALIGNMENT

*Self-Reference* can be used as the basis for an unsupervised-learning loss, that allows aligning instances with specific subspaces that we care about. For this, we propose *Head2Feat*, a method that operates across all attention heads simultaneously and leverages the *self-representation* property to identify regularities across prompts and discover attribute subspaces that capture semantically meaningful directions. Given two sets of attention head outputs, $H^I \in \mathbb{R}^{N \times D}$—the instance vectors we want to evaluate—and $H^C \in \mathbb{R}^{N \times D}$—the attribute-related class vectors—our model seeks to align their representations their shared prototypes.

Our architecture (Figure 4) identifies the subset of attention heads that most effectively encode semantic information by randomly pairing a class vector—from a set of class vectors related to the target attribute subspace—with an instance vector via an attention mechanism and forcing them to always output the same attention weight distribution. Both vectors are projected into a shared attribute space using a common value transformation, enabling meaningful comparisons. Within this space, the model learns a set of prototypes that are encouraged to align with the class vectors, while instance vectors are trained to align with their nearest prototype. This process facilitates the discovery of attention heads that capture semantically relevant features without requiring supervision.

## 3.4 PROJECTION INTO A SHARED LATENT SUBSPACE

Each attention head $H_n$ is independently projected via a head-specific function $f_n$ and then normalized, ensuring all projections reside within a shared latent subspace:

$$\dot{H}_n = f_n(H_n) \tag{3}$$

We then compute the similarity between class and instance vectors using an attention mechanism. The class outputs $\dot{H}^C$ are projected into key space, while the instance outputs $\dot{H}^I$ are projected into the query space. A shared value projection $W^V$ is applied to both:

$$Q_s = \dot{H}^I W_s^Q \quad K = \dot{H}^C W^K \quad V_s^I = \dot{H}^I W_s^V \quad V_s^C = \dot{H}^C W_s^V \tag{4}$$

Due to the need for positive augmentations, we employ several slots that capture the same features but with different combination of attentions. Each slot $s$ uses separate projection matrices $W_s^Q$ and $W_s^V$ while sharing a common $W^K$.

## 3.5 ATTENTION MATCHING

We compute attention interactions only between corresponding heads (i.e., row-wise) from $\dot{H}^C$ and $\dot{H}^I$, focusing on their shared informative content rather than inter-head relations. The attention weights are defined as:

$$A_i = \sum_{j=1}^{n} \dot{H}_{ij}^C \dot{H}_{ij}^I, \quad \text{Attn}(A, V) = \text{softmax}\left(\frac{A}{\sqrt{d_k}}\right) V \tag{5}$$

We enforce that each instance-class pair maintains the same attention distribution, enabling the discovery of a shared attribute space. This is achieved by minimizing the Jensen-Shannon Divergence (JSD) between their attention distributions (Menéndez et al., 1997):

$$\mathcal{L}_{\text{attention}} = \text{JSD}(\text{Attn}_a \parallel \text{Attn}_b) \tag{6}$$

## 3.6 PROTOTYPE LEARNING

To further organize the latent space, we adopt learnable prototypes following Caron et al. (2021). Each learnable prototype represents the model's approximation of the true class vector prototype. We optimize these prototypes by aligning them closely with their corresponding class vectors. This alignment is enforced using a standard cross-entropy loss:

$$\mathcal{L}_{\text{prototype}} = -\sum_{i=1}^{C} y_i \log(\hat{y}_i) \tag{7}$$

## 3.7 SINKHORN NORMALIZATION AND CONTRASTIVE SOFT LABELS

In addition to hard alignment with class prototypes, we apply a soft labeling strategy based on the Sinkhorn-Knopp algorithm (Cuturi, 2013). The normalized prototype assignments serve as target distributions for instance embeddings:

$$\mathbf{P} = \text{diag}(\mathbf{u}) \mathbf{K} \, \text{diag}(\mathbf{v}) \tag{8}$$

The instance loss is defined as the symmetric KL-Divergence between the outputs of different slots computed from the various attention head outputs:

$$\mathcal{L}_{\text{instance}} = \frac{1}{2} \sum_{i=1}^{2} \left[ -\sum_{k=1}^{K} p_k^{(j)} \log q_k^{(i)} \right] \tag{9}$$

### 3.8 FINAL OBJECTIVE

The total loss combines all the components above:

$$\mathcal{L}_{\text{total}} = \lambda_1 \mathcal{L}_{\text{attention}} + \lambda_2 \mathcal{L}_{\text{instance}} + \lambda_3 \mathcal{L}_{\text{prototype}} \tag{10}$$

This framework enables the model to uncover the shared attribute space, allowing each instance vector to naturally align with the most appropriate learned prototype—ideally positioned near its corresponding true prototype.

Through an ablation study of the different losses, we found that all of them are strictly necessary; removing any of them causes the model's accuracy to become essentially random.

For hyperparameter optimization, we performed a grid search over values 0.01, 0.1, 1, and 10 for the three $\lambda$ parameters. The optimal combination was found when all three $\lambda$ values were set to 1.

### 3.9 DATASET

To evaluate the generalization ability of our model and the emergence of semantic prototypes, we tested it on a diverse set of datasets spanning multiple domains. Each dataset consists of **a list of texts and a separate list of classes**, with no knowledge between them, including: country–continent, emotional text–emotion label, animal–biological class, multilingual text–language name, iconic quotes–fictional character, and book excerpts–author name. We include benchmarks such as the XQuAD dataset (Dumitrescu et al., 2021), the Emotion Cause dataset (Ghazi et al., 2015) and TruthfulQA (Lin et al., 2021) among these ones; and some curated prompts to test the results of the activation patching, with facts and text generation. Additional details on these benchmarks are provided in Appendix A.

### 3.10 UNSUPERVISED CLASSIFICATION & DATA CLUSTERING

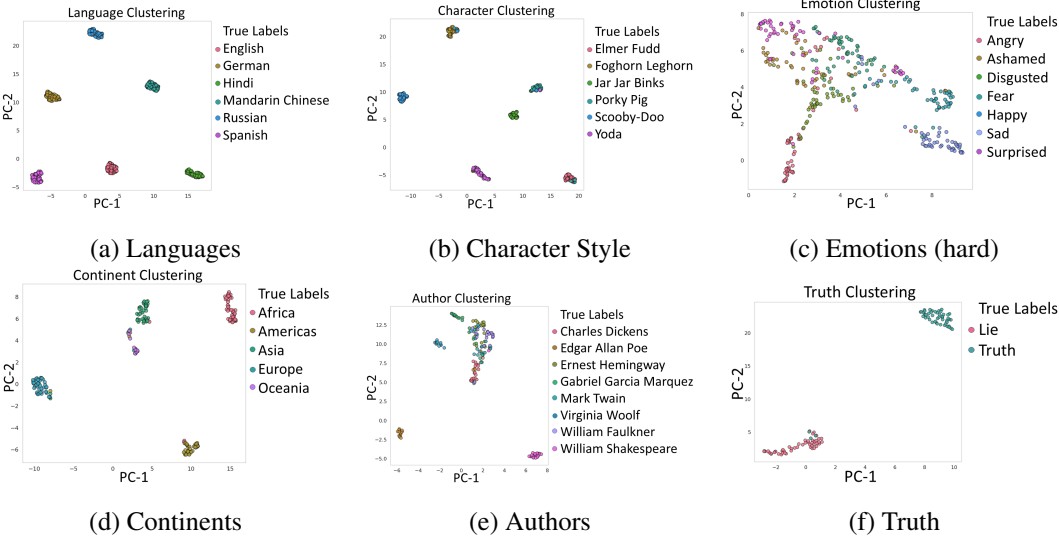

(a) Languages      (b) Character Style      (c) Emotions (hard)

(d) Continents      (e) Authors      (f) Truth

Figure 5: UMAP clusters of different datasets

We evaluated the clusters produced by *Head2Feat* using the Adjusted Rand Index (ARI) (Hubert & Arabie, 1985), Silhouette Score (Rousseeuw, 1987), and classification accuracy based on both the ground-truth labels and the LLM's own predictions. For both *Head2Feat* and *IPA*, classification was performed by assigning each instance to its nearest prototype and comparing the resulting label with the true label.

To provide qualitative insights, we include several visualizations: UMAP projections (McInnes et al., 2018) of the learned representations, alignment heatmaps between prototypes and class labels, and Principal Component Analysis (PCA) of the alignment vectors.

| Data Description | Name | Acc (%) | ARI | Silhouette |
|---|---|---|---|---|
| Countries / Continents | LLM Answers | 100 | - | - |
| | Linear Probe | 93.2 | - | - |
| | *IPA* | 94 | - | - |
| | *Head2Feat* | 94 | 0.78 | 0.71 |
| Emotions | LLM Answers | 68.9 | - | - |
| | Linear Probe | 78.86 | - | - |
| | *IPA* | 58.6 | - | - |
| | *Head2Feat* | 62.6 | 0.38 | 0.42 |
| Animals / Families | LLM Answers | 94.3 | - | - |
| | Linear Probe | 92 | - | - |
| | *IPA* | 72.3 | - | - |
| | *Head2Feat* | 83 | 0.61 | 0.76 |
| Languages | LLM Answers | 100 | - | - |
| | Linear Probe | 100 | - | - |
| | *IPA* | 87.2 | - | - |
| | *Head2Feat* | 100 | 1 | 0.93 |
| Character Style | LLM Answers | 73.6 | - | - |
| | Linear Probe | 87.3 | - | - |
| | *IPA* | 55 | - | - |
| | *Head2Feat* | 87 | 0.71 | 0.87 |
| Authors | LLM Answers | 65 | - | - |
| | Linear Probe | 66.25 | - | - |
| | *IPA* | 61.3 | - | - |
| | *Head2Feat* | 70.6 | 0.46 | 0.54 |

Table 2: Classification accuracy and clustering quality across various semantic attributes. We compare the performance of the LLM, *Head2Feat*, and *IPA* on tasks ranging from token-level features (e.g., languages) to abstract ones (e.g., authorship and truthfulness). For *Head2Feat*, we also report clustering metrics: Adjusted Rand Index (ARI) and Silhouette score.

As shown in Table 2, we bench-marked *Head2Feat* and *IPA* across a diverse set of datasets encompassing a variety of attribute types, including geographic origin, speech patterns, and writing style. While *IPA* performed good in most settings—highlighting the salience of certain semantic attributes—the unsupervised classification approach consistently outperformed it, while obtaining a similar level to the outputs from the LLM. We believe the difference between the methods is mostly related to the difference of obtaining information from all the attention heads, instead of just a single one.

Figure 5 illustrates that for simpler attributes, such as country or profession, the UMAP representations form well-separated clusters, indicating successful prototype assignment. In contrast, for more complex or abstract styles, the cluster boundaries become less distinct, reflecting the complexity of the attribute, and its diminishing shared space between them.

Importantly, two of the tasks in Table 2—Author and Character Style—require abstraction that go beyond surface-level or token-specific cues. In these two, *Head2Feat* obtained better results than the LLM's own predictions, demonstrating its ability to capture higher-level stylistic and discourse features.

## 4 DISCUSSION

Our experiments reveal semantic self-representation in transformers: interpretable features are encoded as stable directional vectors within attention head outputs. We demonstrate zero-shot identification of semantically relevant attention heads through contrastive alignment, revealing which heads encode shared versus class-specific attributes. These findings show that both factual and stylistic information are encoded as distinct prototype vectors that can be isolated without supervision. This prototype invariance—where semantic directions remain stable across contexts—helps explain transformers' generalization abilities, paralleling findings in in-context learning (Hendel et al., 2023). By identifying geometric conditions under which semantic prototypes emerge in an unsupervised manner, we provide new insights into the latent structure governing language generation. This framework enables controllable generation through discovered prototype directions, allowing precise manipulation of semantic attributes without task-specific fine-tuning.

ETHICS STATEMENT

In the preparation of this manuscript, we used OpenAI's ChatGPT (GPT-4) as a language assistance tool to improve clarity, grammar, and readability in parts of the text. All scientific content, ideas, and analyses presented in this paper are the original work of the authors. The use of ChatGPT was limited to language refinement and did not influence the experimental design, results, or conclusions.

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

## APPENDIX AND SUPPLEMENTARY MATERIAL

The code and datasets used in our experiments will be released upon acceptance of the paper. A link to the repository will be included in the camera-ready version.

## A    DATASETS

We used and constructed several small datasets to observe where all of this attributes are encoded in LLMs and if all semantic features work as prototypes.

**Famous Locations and People From Same Country**    We formed two datasets consisting of 100 sentences each one related to the description of famous monuments and the other for famous people, from 5 different countries and we extracted the final 10 attention head outputs from it.

**Languages**    To evaluate the ability of *Head2Feat* to distinguish between languages, we employed the XQuAD (Dumitrescu et al., 2021) dataset a benchmark for cross-lingual question answering. We sampled 200 questions per language, covering English, Spanish, Russian, Hindi, German, and Mandarin Chinese. This setup enables us to test whether language identity can be reliably inferred from the internal representations of the model.

**Emotions**    To investigate the representation of affective states, we used two datasets targeting emotional content. First, we constructed a controlled dataset of 100 English sentences, each expressing one of five emotions—joy, sadness, anger, fear, and disgust—without explicitly naming the emotion in the text.

Second, we used a more challenging benchmark: a subset of the Emotion Cause dataset (Ghazi et al., 2015), which includes 1,594 English sentences annotated with seven emotion labels (fear, sadness, anger, happiness, surprise, disgust, and shame). We sampled 600 random examples from this dataset for training and validation purposes.

**Famous Fictional Characters**    In addition, to probe where stylistic and character-specific features are encoded in the model, we constructed a small-scale dataset centered on fictional characters with distinctive linguistic patterns. For each of six well-known characters—Elmer Fudd, Foghorn Leghorn, Jar Jar Binks, Porky Pig, Scooby-Doo, and Yoda—we collected 50 iconic phrases from publicly available sources such as fan wikis and quote databases.

**Literature Authors**    We obtained 20 book quotes per literary author—William Faulkner, Gabriel García Márquez, Ernest Hemingway, Edgar Allan Poe, Virginia Woolf, William Shakespeare, and Mark Twain—from the website Goodreads.

**True or False Statements** We compiled a dataset of 100 sentences, evenly split between true and false statements. Several of the false statements were constructed as direct negations or opposites of their true counterparts.

Finally, to study the encoding of categorical features in more controlled settings, we compiled three structured datasets.

**Countries** The first contains the names of all recognized countries and a predefined set of continent classes (Africa, Asia, Europe, Oceania, and the Americas), used to probe how geographic categories are internally represented.

**Animals** The second includes a collection of animal species names and a set of biological class categories—mammals, invertebrates, birds, amphibians, reptiles, and fishes—used to investigate how categorical distinctions among animals are encoded.

**Famous People / Jobs / Countries** The final dataset consists of the names of 100 famous individuals, each of whom can be categorized along two dimensions: country of origin and occupation. The dataset includes 20 examples for each of five countries (USA, Japan, Brazil, India, and France) and five occupations (athlete, scientist, politician, musician, and actor), allowing us to evaluate how different clustering objectives emerge depending on the targeted feature.

## B ATTENTION HEAD WEIGHTS

To better isolate the features of interest, we incorporated positive augmentations, following the approach commonly used in contrastive learning. Specifically, we extracted multiple output vectors from our attention mechanism, where each query-value pair—referred to as a slot in our design—is orthogonal to the others. We employed two slots, yielding two distinct attention weight distributions per experiment.

In some cases (e.g., Figures 13a and 13b), the attention was sharply focused on a single head, suggesting a localized and interpretable signal. In contrast, other examples (e.g., Figures 11a and 11b) exhibited more diffuse attention distributions, with no clear subset of heads responsible for encoding the relevant prototypes. Across most scenarios, the key difference between the slot-specific distributions was the relative weight assigned to particular heads, rather than a change in which heads were active.

## C PROBABILITY HEATMAPS

We visualize the alignment of individual instances across all classes in the dataset using heatmaps. In most cases, the distributions are sharply concentrated within the correct class, indicating strong alignment. However, for more nuanced datasets such as Emotion-Easy (Figure 15) and Emotion-Hard (Figure 16), the class separability is less pronounced, despite high classification accuracy. This can be attributed to the inherently overlapping nature of emotional expressions, which often encode multiple affective cues simultaneously. Our model captures this mixture, typically assigning high weight to a dominant emotion while also registering lower intensities for secondary ones. For instance, in Emotion-Easy, instances labeled as disgust frequently show secondary associations with anger or sadness, and fear often co-occurs with disgust. Interestingly, these co-occurrence patterns are asymmetric and not always bidirectional.

## D PROBE'S WEIGHTS CONVERGENCE TO PROMPT OUTPUTS

We trained linear probes to classify each dataset's classes using attention head outputs. For each probe, we identified the attention heads with the highest similarity to the class prompt representations

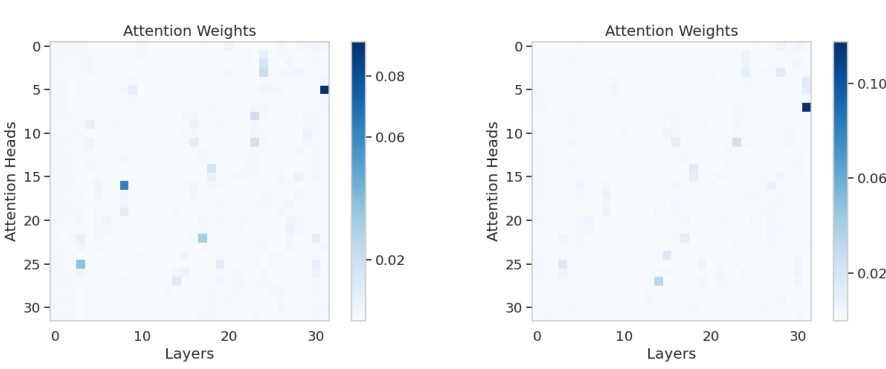

(a) Attention Head Weight Distribution, Slot 1  (b) Attention Head Weight Distribution, Slot 1

Figure 6: Continents

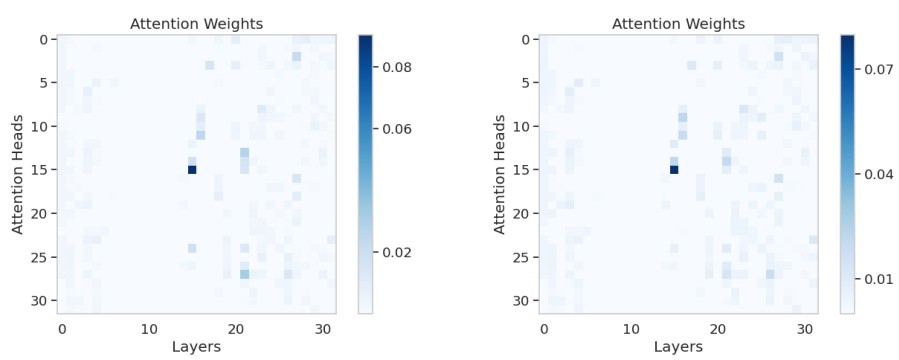

(a) Attention Head Weight Distribution, Slot 1  (b) Attention Head Weight Distribution, Slot 1

Figure 7: Animals

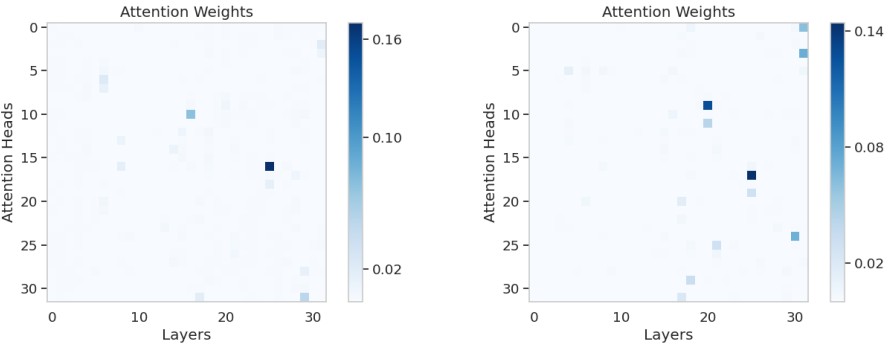

(a) Attention Head Weight Distribution, Slot 1  (b) Attention Head Weight Distribution, Slot 1

Figure 8: Emotions (easy)

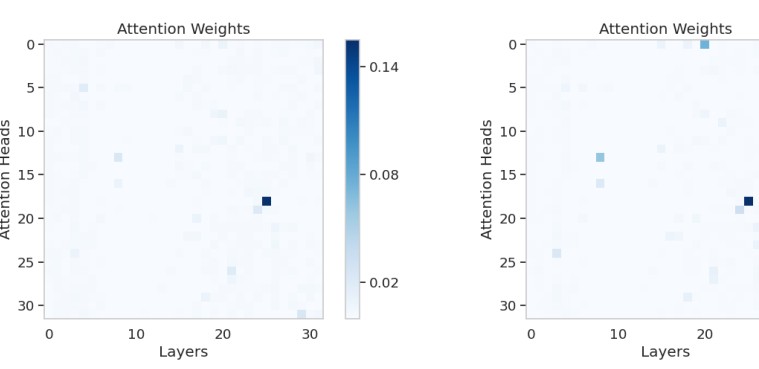

(a) Attention Head Weight Distribution, Slot 1    (b) Attention Head Weight Distribution, Slot 1

Figure 9: Emotions (hard)

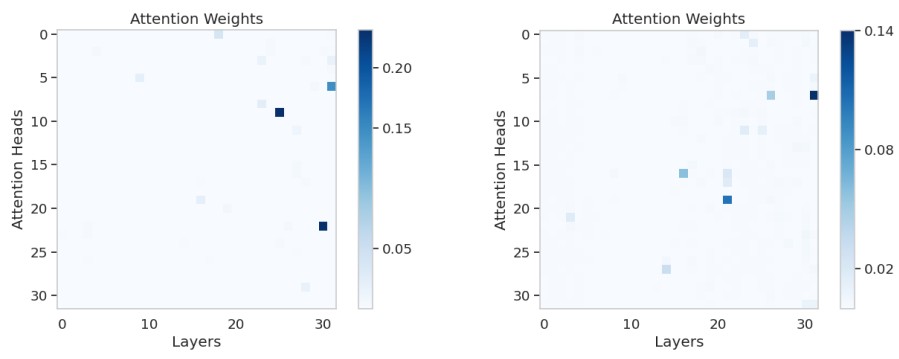

(a) Attention Head Weight Distribution, Slot 1    (b) Attention Head Weight Distribution, Slot 1

Figure 10: Languages

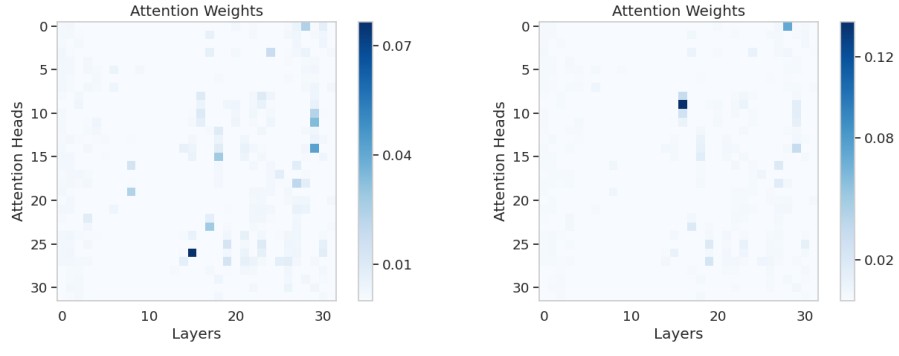

(a) Attention Head Weight Distribution, Slot 1    (b) Attention Head Weight Distribution, Slot 1

Figure 11: Authors

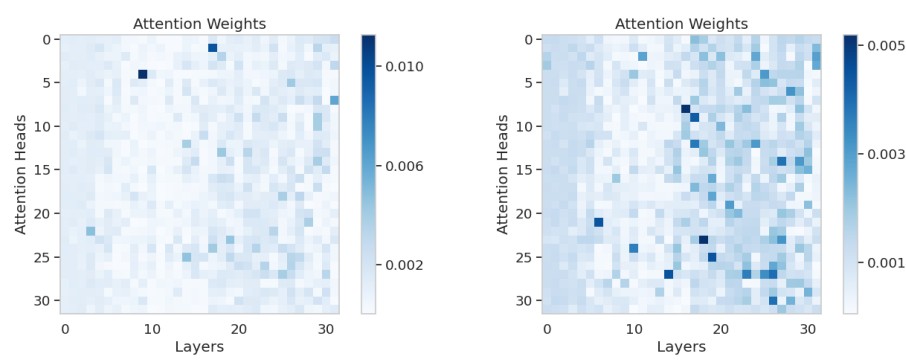

(a) Attention Head Weight Distribution, Slot 1    (b) Attention Head Weight Distribution, Slot 1

Figure 12: Characters

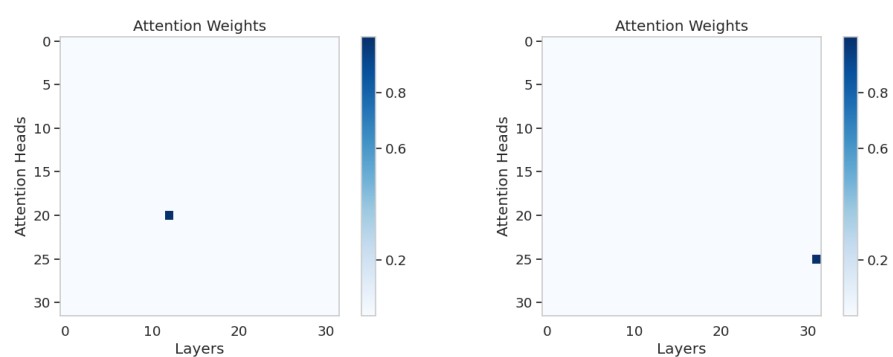

(a) Attention Head Weight Distribution, Slot 1    (b) Attention Head Weight Distribution, Slot 1

Figure 13: Truth

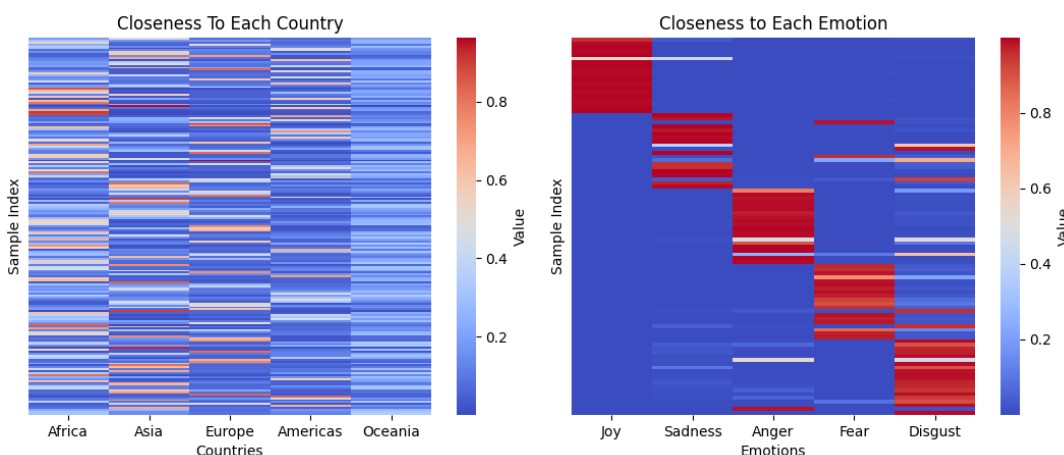

Figure 14: Continents                    Figure 15: Emotions (easy)

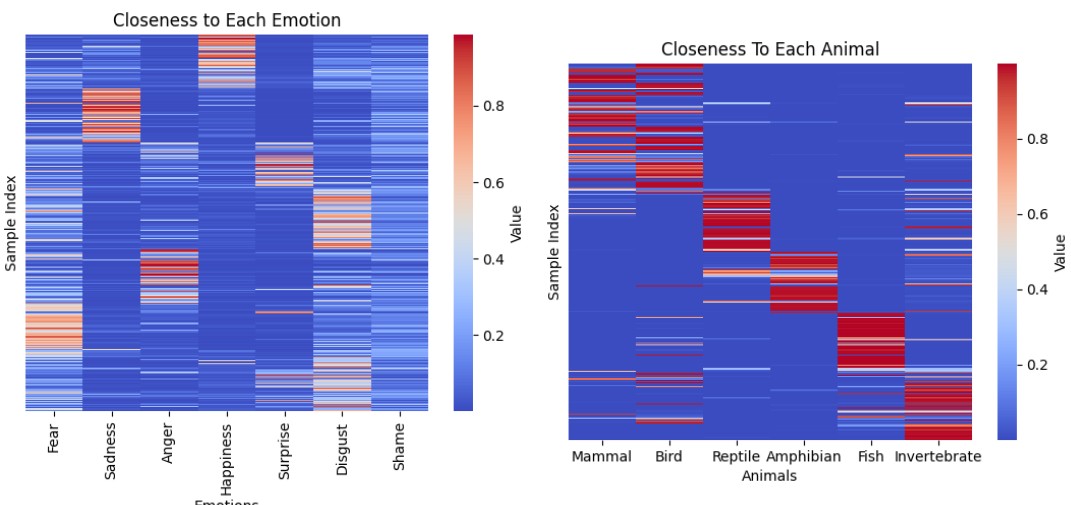

Figure 16: Emotions (hard)

Figure 17: Animals

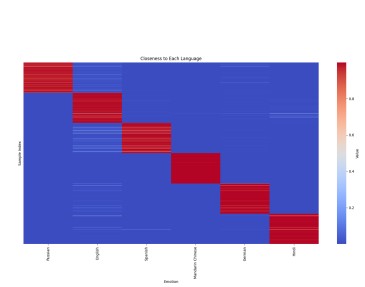

Figure 18: Languages

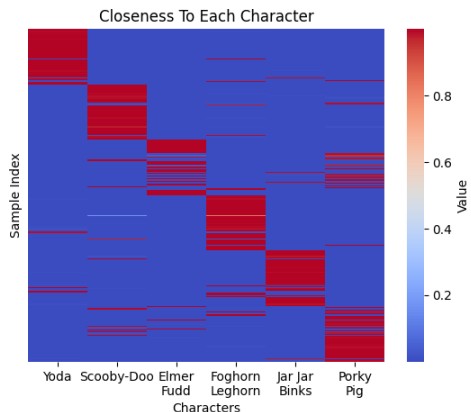

Figure 19: Characters

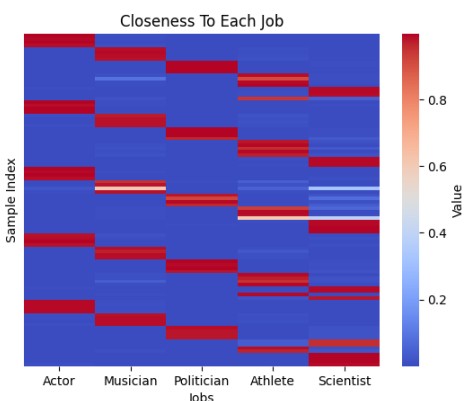

Figure 20: People / Jobs

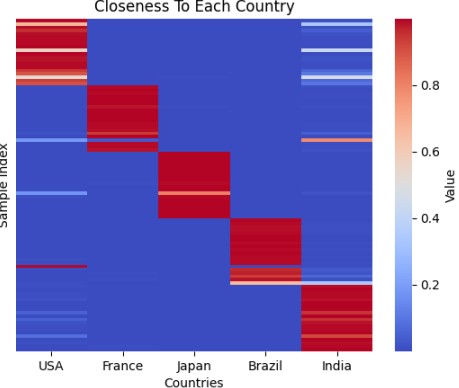

Figure 21: People / Countries

and those achieving the highest classification accuracy. Across most datasets, the top-performing probes exhibited a consistent upward trend in similarity over the course of training, with relatively low variance, indicating stable convergence. Their most influential attention heads aligned strongly with the target class prompts. An exception was the countries dataset, where no attention heads showed a significant correlation with the prompts. Similarity scores varied across datasets, with an average around 0.5.

# E  CLUSTERS

We visualize the clustering results of our various datasets using UMAP and PCA. In most cases, the clusters exhibit clear separability between classes. For more challenging datasets—such as Authors (Figure 38), Characters (Figure 39), and the Hard Emotion subset (Figure 36)—the most distinct classes remain well-separated, though some classes exhibit significant overlap and cannot be completely disentangled.

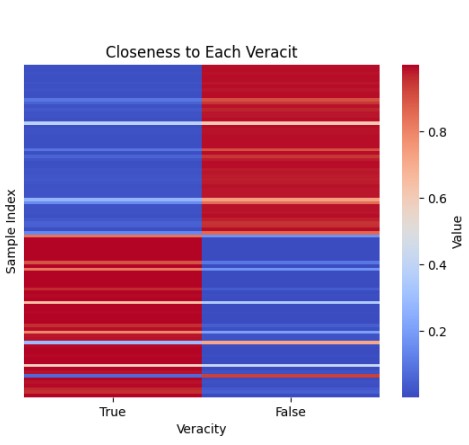

Figure 22: Truth

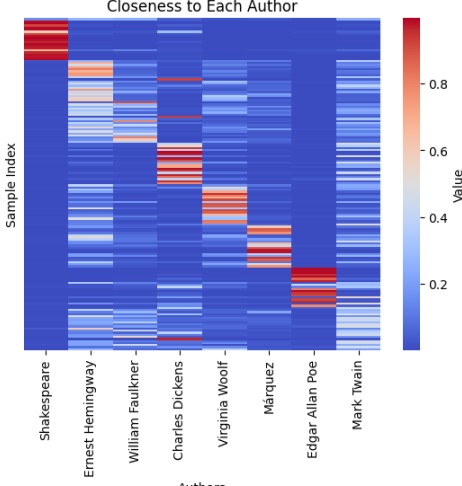

Figure 23: Authors

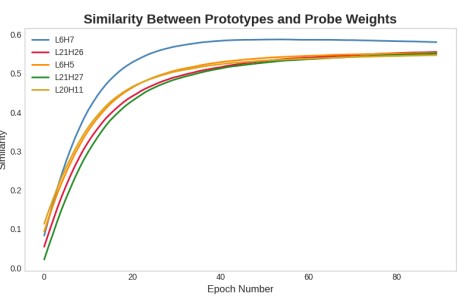

Figure 24: Emotions (hard)

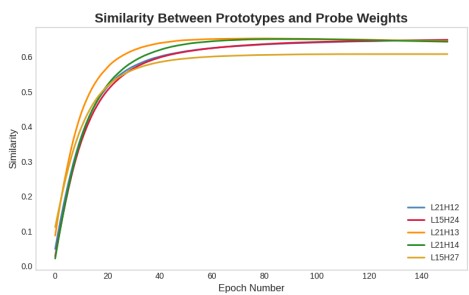

Figure 25: Animals

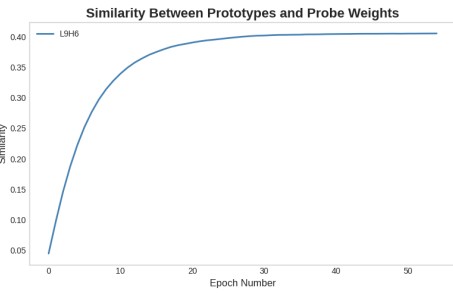

Figure 26: Languages

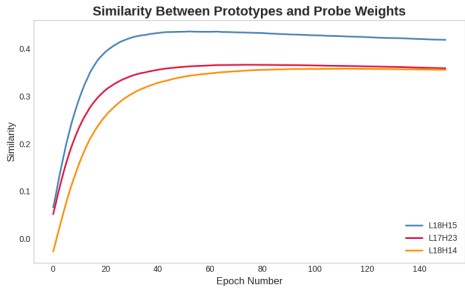

Figure 27: Characters

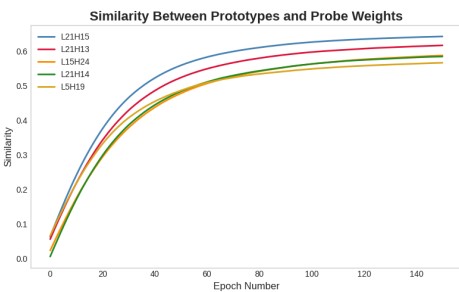

Figure 28: People / Jobs

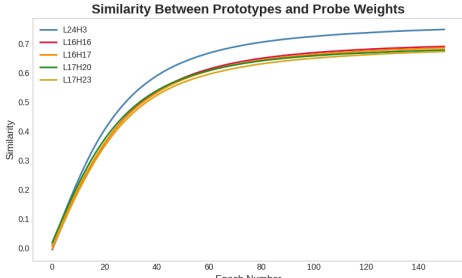

Figure 29: People / Countries

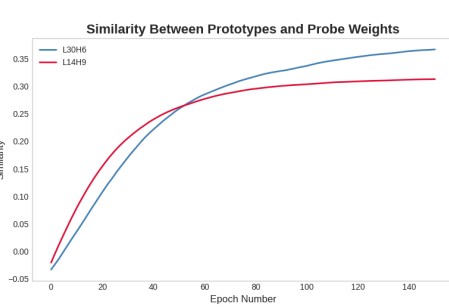

Figure 30: Truth

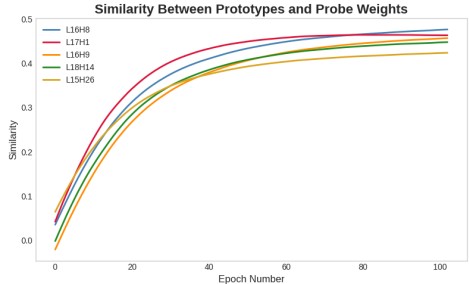

Figure 31: Authors

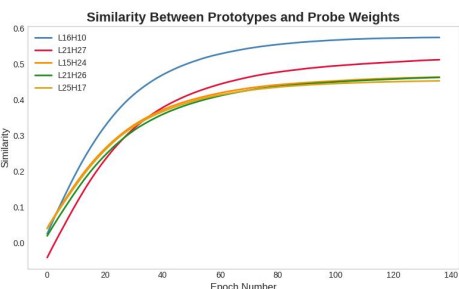

Figure 32: Emotions (easy)

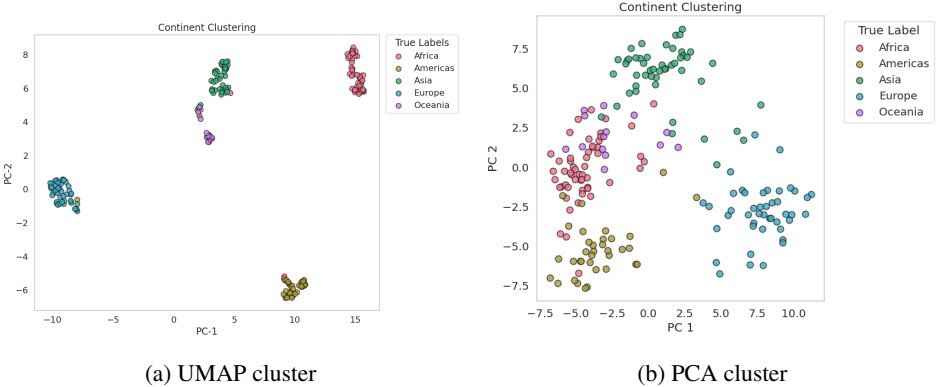

(a) UMAP cluster

(b) PCA cluster

Figure 33: Continent Clusters

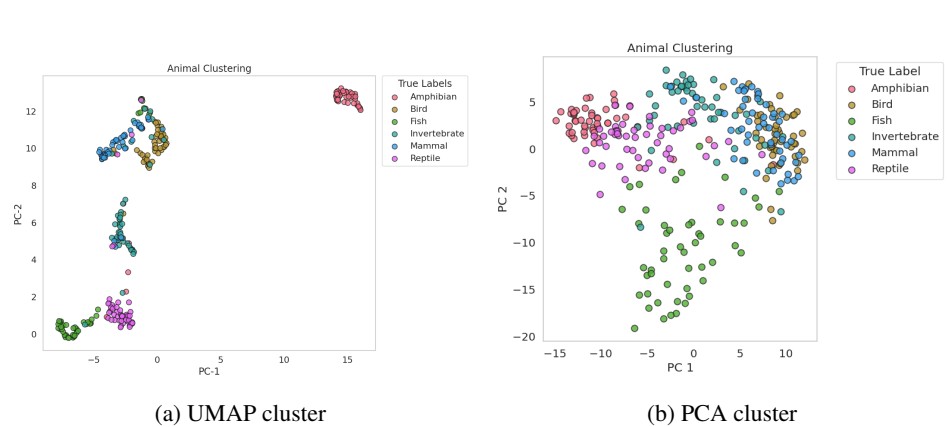

(a) UMAP cluster                    (b) PCA cluster

Figure 34: Animal Clusters

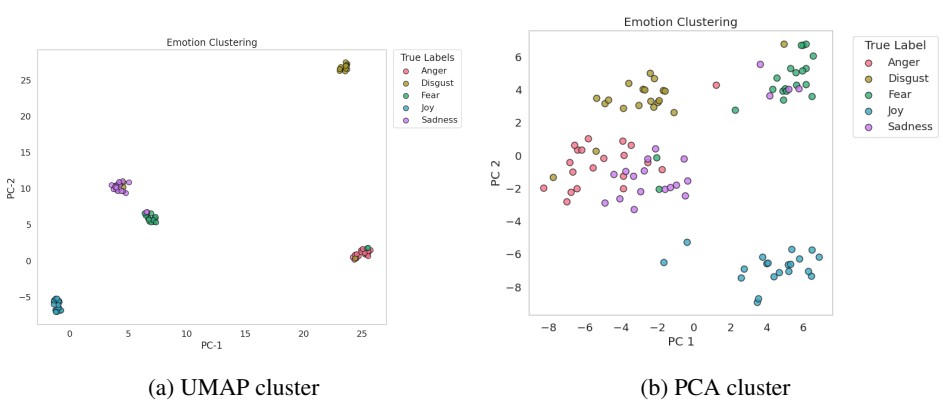

(a) UMAP cluster                    (b) PCA cluster

Figure 35: Emotion (easy) Clusters

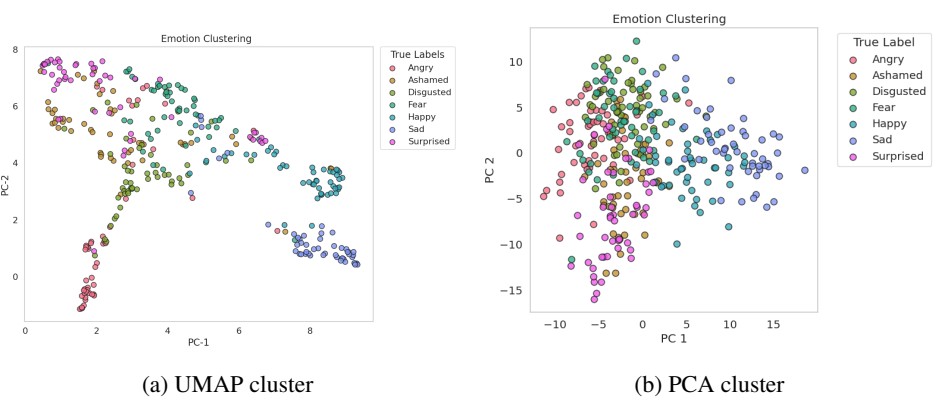

(a) UMAP cluster                    (b) PCA cluster

Figure 36: Emotion (hard) Clusters

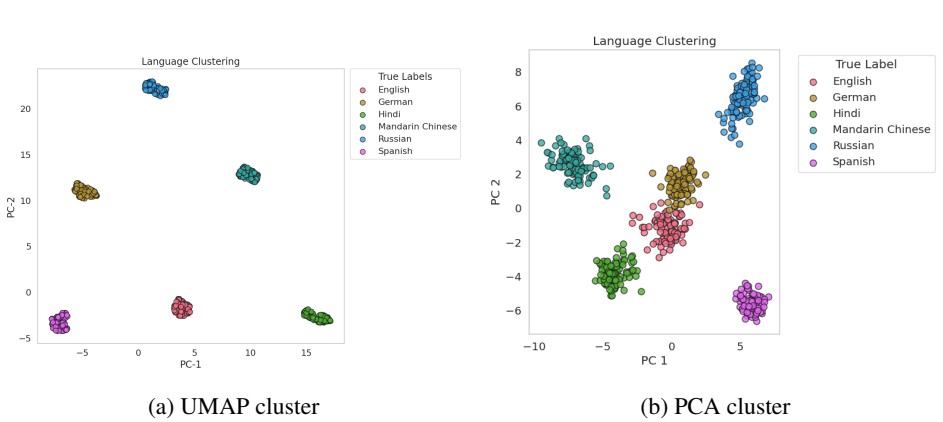

(a) UMAP cluster
(b) PCA cluster

Figure 37: Language Clusters

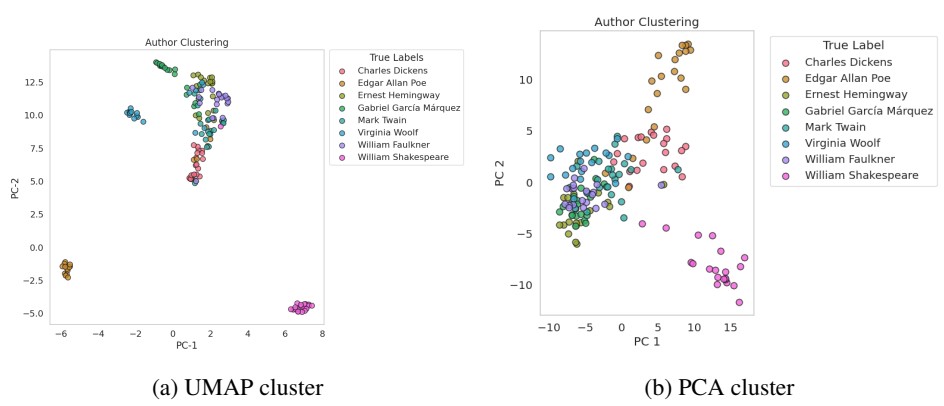

(a) UMAP cluster
(b) PCA cluster

Figure 38: Author Clusters

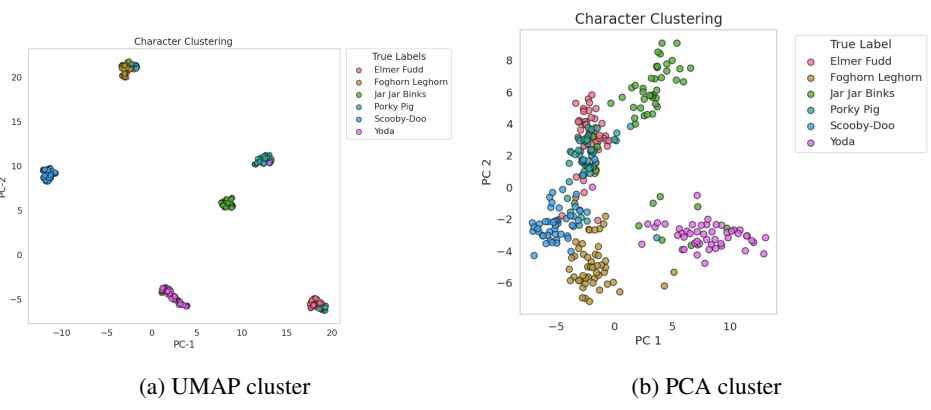

(a) UMAP cluster
(b) PCA cluster

Figure 39: Character Clusters

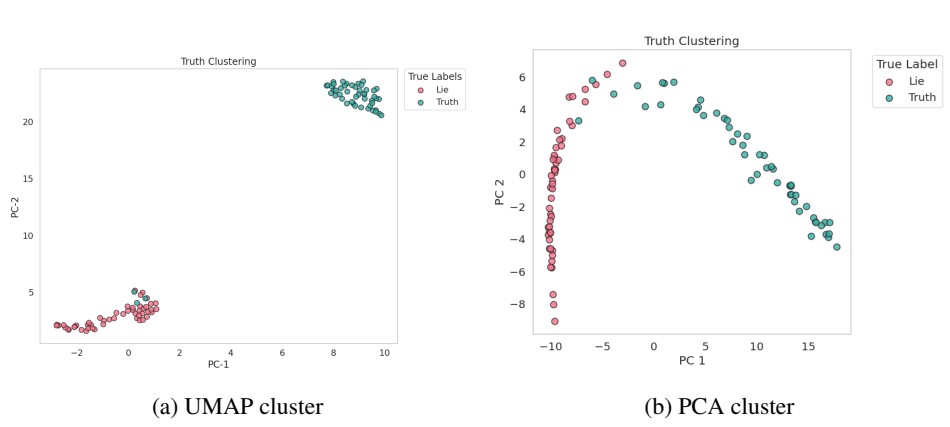

(a) UMAP cluster

(b) PCA cluster

Figure 40: Truth Clusters

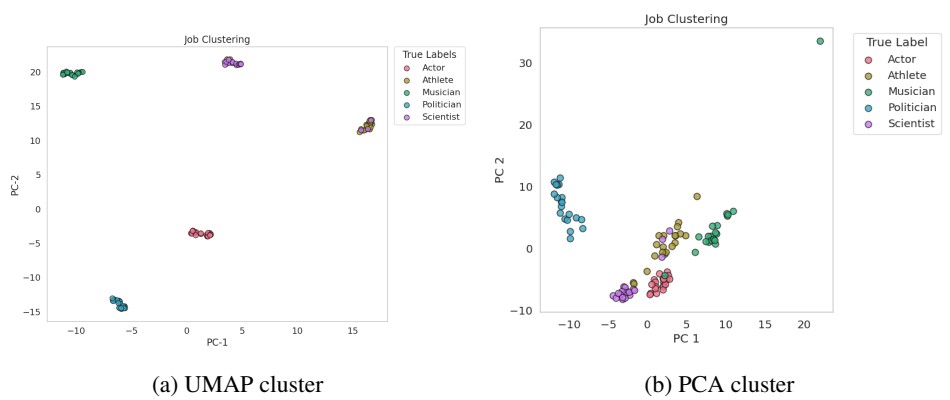

(a) UMAP cluster

(b) PCA cluster

Figure 41: People-Job Clusters

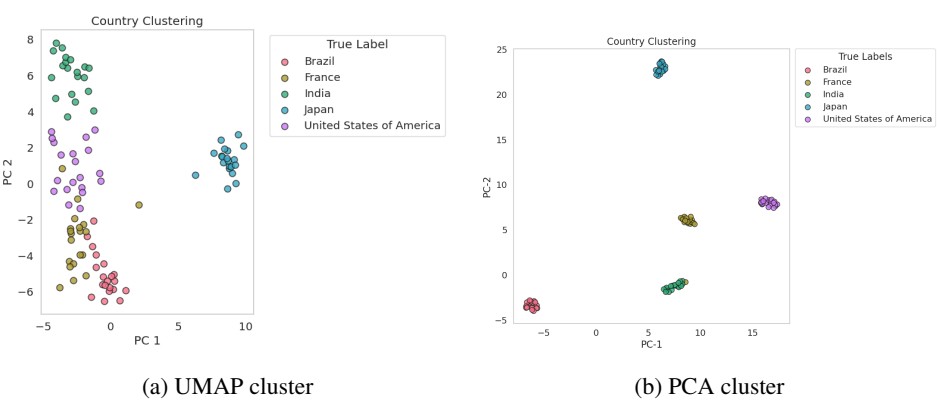

(a) UMAP cluster

(b) PCA cluster

Figure 42: People-Country Clusters

