# OpenReview forum: "Identity-Projection As A Way of Analyzing Attention Heads In Transformers"
_ICLR.cc/2026/Conference — ICLR 2026 Conference Withdrawn Submission_

### Official Review · Reviewer_vFJs · 2025-10-26

**Soundness:** 1
**Presentation:** 1
**Contribution:** 2
**Rating:** 0
**Confidence:** 4

**Summary:**

This paper introduces Identity-Projection, a geometric hypothesis that semantic concepts in transformers persist as invariant prototype directions, and evince themselves to different degrees in model representations across contexts depending on the semantic relevance. The paper develops two methods—IPA (zero-shot head attribution) and Head2Feat (unsupervised prototype alignment)—to identify attention heads encoding specific features. The authors evaluate these methods on multiple tasks settings, domains, and demonstrate their advantage over the traditional supervised methods of training linear probes on attention heads activations.

**Strengths:**

The idea of using the alignment between head activations and paradigmatic reference concept directions extracted using the concept tokens themselves is novel. The experiments regarding the correlation between the heads with the highest influences scores for promtps with explicit or implicit reference to the concept are interesting.

**Weaknesses:**

The paper contains numerous serious flaws in the **description of its proposed methods** and the **presentation of its results**, which I list below.

---

**Identity Projection and Influence Score**

**Methodological Issues**: The authors define the prototype direction of a token (or concept) $t$ to be the activation of a head on an input consisting solely of that token. However, since this direction varies across heads, it is unclear how it can be treated as an invariant representation of the concept in the model's representation space. The resulting influence score, which measures the cosine similarity between a head’s activation $v$ on a general input and the prototype direction, is therefore conceptually unsound due to the variability of the prototype direction. In the following experiment, the authors extract the prototype direction for the contrast between “France” and “Spain” using the mean difference of the activation of some head $v_i$ over two groups of prompts. However, they do not specify which head was used, nor what kinds of prompts were employed (e.g., “France” and “Spain” alone? Prompts from the explicit/implicit dataset?), which causes great confusion and renders the results difficult to interpret.

**Presentation Issues**: Figure 1(b) uses class-averaged cosine similarities as its y-axis, whose values should range between -1 and 1. However, the figure reports values as high as 3.5, which is mathematically impossible.

---

**Section 3.1 and 3.2**

**Methodological Issues**: One of the paper’s central claims is that it develops alternative methods to traditional linear-classifier-based approaches for identifying heads encoding specific semantic concepts. However, in the head visualization experiments in 3.1 and the activation patching experiments in 3.2, the authors explicitly use F1-scores based on trained linear probes to identify influential heads. This contradicts their stated goal and creates genuine confusion about the novelty of their method.

**Presentation Issues**: The caption of Figure 2 is incorrect, as it repeats that of Figure 3. Lines 258–261 and 263–267 reiterate the same idea. The meaning of each column in Table 1 is not explained and is deeply confusing. Several sentences appear entirely unrelated to the surrounding context, such as lines 222–223. Moreover, some experiment results are referenced but never shown—for example, lines 259–261 discuss the significance of the “English” concept in activation patching, yet no corresponding figure or table is provided. The cosine similarity values in Figure 3 exceed 1.5, which contradicts the mathematical definition of cosine similarity.

---

**Head2Feat**

**Methodological Issues**: The presentation of the “Head2Feat” method in Sections 3.3–3.8 is nearly incomprehensible. Numerous concepts are introduced without explanation; many variables are defined but never used; and several others are used without ever being defined. To name a few examples: How are $H^{I}$ and $H^{C}$ in 3.3 obtained, and from what inputs? The query and embedding matrices with slot subscripts in 3.4 are defined but never referenced again, and no information is provided on how matrices across different slots are integrated. The key matrix is introduced but never used in subsequent derivations. In 3.5, which value matrix does $V$ refer to? What exactly are $Attn_a$ and $Attn_b$? What are the $y_i$ and $\hat{y_i}$ used to compute $L_{prototype}$ in 3.6? What do $u$, $v$, $K$, $P$, $p_k$, and $q_k$ represent in 3.7? After reading these sections, it is impossible to understand how to implement the Head2Feat approach.

**Presentation Issues**: In 3.8, the authors briefly mention the conclusions of the ablation studies, yet the actual results of these studies—such as those exploring different loss combinations and hyperparameter choices—are not presented anywhere in the paper.

**Questions:**

Please see weaknesses.

---

### Official Review · Reviewer_L6vA · 2025-10-28

**Soundness:** 3
**Presentation:** 1
**Contribution:** 3
**Rating:** 2
**Confidence:** 3

**Summary:**

The paper presented a new approach, IPA (Identity Projection Analysis), to identify the influential attention heads for a given text concept. It further proposed unsupervised classification approach based on IPA. Despite that the approaches seem novel, the lack of clarity of the paper makes it hard to assess the paper.

**Strengths:**

The proposed method is novel and can be potentially influential to the field by having simple ways to identify salient attention heads. It could benefit a broad range of applied ML research that uses transformer as backbone.

**Weaknesses:**

The lack of clarity of the paper makes it hard for me to access the paper. I cannot fully understand the method proposed and need to make guesses throughout reading the paper. The following listed my concerns about clarity that I hope the authors can resolve:

- It is very confusing to me that you introduced in Proposition 1 the property of Self-consistency, Distance-decay, and Orthogonality, but they are not used in the paper afterwards. Isn't it suffice to just define the Prototype vector? I am not sure where you need the remaining properties. Also, I did not find proofs for why, and under what assumption does these properties holds. It doesn't look trivial to me. The authors should revise Proposition 1 and make clear the role of it in the paper.

- On Equation 1, what is the input vector $v$? It reads on line 129: "Then, to quantify the influence of a prototype on a given prompt, we project the input vector $v$ onto the prototype vector $p_t$". I assume the input vector $v$ is related to the prompt, as $p_t$ does not encode the context but only the token $t$. However, I did not find any formal definition of $v$ or even its dimension, how do you extract the vector $v$ out of the potentially variable length prompt?

- Throughout the paper, is there any instances where you use $h_n(t,C)$ instead of $h_n(t,t)$ other than in proposition 1?

- It is also confusing to me on how to do the Contrastive token analysis. It says you use Equation 2 to calculate the prototype direction in $\mathbf{p}_c$. I assume this is used to replace the $\mathbf{p}_t$ in Equation 1 for calculation. If so, where does the attention head involves in the calculation? as the original $p_t = h_n(t, t)$ is where the attention head involves. Or does the $\mathbf{p}_c$ actually replaces the $v$ in Equation 1? This part I cannot understand. The authors should explain this more carefully.

- I am struggling to understand what is the advantage of the proposed Head2Feat. I am not sure it can be called unsupervised, as the model do use use the class vectors during training. Also, what is the advantage of this method over linear probing? in Table 2, I don't see much a difference on performance. Additionally, the ARI and Silhouette scores lack a reference baseline, making it hard to interpret these scores.


Typo:
- Line 162: promtps
- Line 191: All the heatmap graphs can be found in the Appendix -> Should reference to D
- Line 361:  align their representations their shared prototypes.

I am willing to raise the score if the authors can address the issues.

**Questions:**

See the above weaknesses.

---

### Official Review · Reviewer_vrB6 · 2025-10-28

**Soundness:** 2
**Presentation:** 2
**Contribution:** 3
**Rating:** 4
**Confidence:** 2

**Summary:**

This paper introduces the concept of "identity-projection", arguing that prototypes such as "France-location" are encoded as stable directions in the latent space of attention heads. In order to find them, the paper presents two methods:

1. Identity Projection Analysis (IPA): A zero-shot attribution method to identify which attention heads encode known semantic concepts.
2. Head2Feat: A novel unsupervised method designed to simultaneously discover latent semantic clusters (like author style) and identify the heads responsible for them.

The authors attempt to validate these methods through causal steering experiments and unsupervised clustering benchmarks, where Head2Feat shows promising results against supervised baselines, such as linear probes.

**Strengths:**

1. The paper addresses an important question: “how do LLMs represent abstract concepts?” They offer an intuitive prototype model to understand it.
2. The paper proposes a novel unsupervised method to find attention heads encoding semanticallty relevant features, with good results in some cases compared with supervised linear probes.

**Weaknesses:**

Weaknesses:
1. The "France/Italy" dataset used to validate Proposition 1 (Figure 1) is not described. The construction of the "explicit, implicit, unrelated" sets is undefined, making the experiment non-reproducible.
2. It is unclear how the F1-score is computed in Section 3.1.
3. I do not understand the methodology of Section 3.2. In standard causal mediation analysis, we need to define the counterfactual input and the metrics to evaluate the success of the interventions. Both are not clear for me in the text. I also do not fully understand the metrics of Table 1. Also, the statement "the metrics showed good results" (line 269) is unclear, with no immediate definition of those metrics. Moreover, line 304 mention that only 4 prompts are needed to obtain the scores, but it is not clear for me how this is generalizable to other prompts.
4. The paper claims an ablation study (Section 3.8) found all Head2Feat loss components "strictly necessary" but presents no data, table, or figure to support this.

Typo/Minor Suggestions:
- Line 70: high instead of hgih
- Proposition 1: $T$ is not defined.
- Equation 2: $\frac{1}{|P|}$ and $\frac{1}{|N|}$
- I recommend to add a citation to “Function Vectors in Large Language Models”: https://openreview.net/pdf?id=AwyxtyMwaG in the paragraph “Model Steering and Activation Engineering”.

**Questions:**

1. Can you please describe the construction of the "France/Italy" dataset? What specific prompts were used for the "explicit, implicit, and unrelated" sets?
2. Could you please clarify the "F1-score" metric used in Figure 2?
3. Could you please define the metrics in Table 1? What do "BERT-Spanish Score" and "accuracy" measure and how are they computed?
4. Can you walk me though the methodology of Section 3.2? What is the counterfactual and original prompts? What is the metric measured after patching from the corrupted activations to the clean? How the effect of the patching is measured?

---

### Official Review · Reviewer_F3ZA · 2025-10-31

**Soundness:** 2
**Presentation:** 1
**Contribution:** 2
**Rating:** 2
**Confidence:** 4

**Summary:**

The manuscript investigates the latent representations produced at the outputs of attention heads in large language models (LLMs) to analyze the functional role of attention in input processing. From these representations, the authors derive _prototypes_—concept-representative vectors used within an identity-projection (IP) framework—to identify when a given concept is activated by contextual stimuli. The IP mechanism is then applied to determine which attention heads encode or respond to specific concepts and to enable efficient activation patching, thereby steering model outputs toward desired themes. Additionally, the same framework is leveraged for automatic feature extraction and attribution, called Head2Feat, by employing sets of class vectors accounting for the categories of interest and instances clustered according to which features are expressed therein.

**Strengths:**

1. The presented method called Identity-Projection is general and seems to work well in all the presented scenarios
2. The Head2Feat algorithm looks an interesting and powerful feature extraction and attribution method

**Weaknesses:**

It is hard, if not impossible, to properly gauge the relevance of the proposed methods, since the models employed for the investigations are nowhere specified and there is no direct comparison with other known methods in terms of performances or computational complexity.  In general, the presentation of the algorithms, especially Head2Feat, is not sufficiently clearly explained and the description of experiments is often superficial or relegated to the appendix (without explicitly mentioning it).

**Questions:**

1. Do I understand correctly that $h_n​(t,C)=\sum_{i\in C} \text{​softmax}(q_t^Tk_i/d_H)v_i$ i.e. the $t$ acts as a query token and $C'$ provides the key/value tokens it attends to and not the opposite? I would find it helpful to be briefly specified in the text, even as a footnote
2. You partially mention it in the last paragraph before section 3.1 ("aligning contrastive prompts with the respective contrastive token"), but can you be more precise on why you carried out the validation experiment​ on a vector representing France-Italy rather than using just the token "France" on the 3 french subsets? I think I get the rationale behind it, i.e. observing the overlap specifically with "France" by differencing out the shared background structure — only the features that distinguish France from Italy survive and not and not just possibly all the other meaning France might be encoding (country, proper noun, location, etc.). I think mentioning this detail would make both the experiment and the reading more clear. The $v_i$ in Eq. 2 are the representations of the last token of the outputs of the attention heads?
3. Procedures are often poorly described:
	- In section 3.1, please refer to appendix for set of sentences used for the experiment of multi-class token classification. concerning the contrastive analysis, what are the two classes or pairs considered from which the prototype is extracted?
	- For the experiment in section 3.2, it is not clear which prompts are being used to generate the continuations. Some examples of inputs, plain output and steered output could help. Concerning Table 1, it is not reported how metrics are computed.
	- sections 3.3 to 3.10 concern the description of H2F, I think the headers layout should be diversified from that of 3.1 and 3.2, where different routines (IPA) are presented. The whole H2F procedure is a bit hard to follow, I find it could be explained more clearly. What's the outcome of the procedure? Which is the latent space in which the clustering procedure is performed?
		- how are the "head-specific functions $f_n$" computed or initialized?
		- terms in Eq 8 and 9 are not explained
		- How expensive is the minimization of the loss in Eq 10? How is it performed? What is the overall architecture employed?
		- Which linear probes are used in experiment of sec 3.10? On which representations? Given the very high performances, close if not superior to LLM answers, I would like the authors to comment on the advantages for their formalism
		- In figure 5, for Umap projection it is not appropriate to name the axes as principal components. Which parameters have been used for the projections?

4. typo at line 70 (hgih -> high). Right panels of Fig 2 and 3 are smaller. Caption of Figure 2 is likely to be wrong. Sub-captions of Figures 6 to 13 are always referring to Slot 1

---

### Note · Authors · 2025-11-15

I have read and agree with the venue's withdrawal policy on behalf of myself and my co-authors.